# ADAPTIVE OFFLINE DATA REPLAY IN OFFLINE-TO-ONLINE REINFORCEMENT LEARNING

## ABSTRACT

Offline-to-online reinforcement learning allows agents to benefit from both sample efficiency and performance by integrating offline and online learning stages. One central challenge in this setting is how to effectively combine the rich experiences collected offline with real-time online explorations. Previous works commonly adopted predetermined mixing ratios for offline data replay as a primary approach to harness offline data in online training. However, determining the best mixing ratio that suits a specific environment and offline dataset often demands empirical adjustments that are context-specific. To address this, we propose a new approach for offline data replay that dynamically adjusts the mixing ratio based on the exploration reward of the agent in the online learning phase. Specifically, our method employs a bandit model to explore and exploit various mixing ratios, subsequently establishing a dynamic adjustment pattern for these ratios to enhance offline data utilization. Empirical results demonstrate that our approach outperforms conventional offline data replay methods, consistently proving its effectiveness across various environments and datasets without the need for targeted, context-specific adjustments.

## 1 INTRODUCTION

Reinforcement learning (RL) has showcased its advantage in a range of applications, achieving notable success in many tasks (Arulkumaran et al., 2017; Mazyavkina et al., 2021; Nakamoto et al., 2023). However, training a policy from scratch in standard online RL settings often demands a vast amount of samples, highlighting an inherent issue of low sample efficiency (Yu, 2018; Zhang et al., 2021). This limitation restricts the practicality of RL, particularly in contexts such as autonomous driving, where interactions with the environment incur significant costs.

Recognizing the abundant offline data available for many real-world tasks, there is growing interest in the offline-to-online RL paradigm (Kostrikov et al., 2021; Lee et al., 2022). Here, agents start by pre-training from offline data and transition to online training (Levine et al., 2020). Yet, the question of how best to utilize this offline data to strengthen online policy learning remains open. Recent works have indicated that under certain conditions, merging offline data into the online training can significantly outperform purely online approaches (Lee et al., 2022; Nakamoto et al., 2023). Common strategies involve populating the online replay buffer with offline data (Vecerik et al., 2017; Hester et al., 2018) or setting pre-determined ratios for mixed offline and online data replay during online training (Kalashnikov et al., 2018; Hansen et al., 2022; Zheng et al., 2023). While the former can be sensitive to offline data quality and quantity, the latter, such as symmetric sampling, tends to be more adaptable in multiple tasks(Ball et al., 2023; Nakamoto et al., 2023).

Nevertheless, the pattern of offline data replay during online training significantly influences agent performance across various contexts. The optimal mixing ratio is not universal – different tasks and environments exhibit divergent performance under various fixed ratios. As an example shown in Figure 1, we deploy an offline-to-online algorithm, calibrated Q-learning (Nakamoto et al., 2023), on two different environments, and we use two different mixing ratios (0.1 and 0.5) for offline data replay to present their online training performance, and it could be seen that 0.1 is better for the "mixed" environment, while 0.5 is better for the "complete" environment. Moreover, static ratios throughout training might not always be the best choice; dynamic adjustments may lead to better learning performance and utilization of offline data in the online phase of offline-to-online RL.

Finding the most effective mixing ratio pattern for specific contexts typically involves empirical and context-specific tuning, due to the lack of a universally robust approach.

In order to optimize the offline data replay in the online training phase, we propose that the mixing ratio should adapt to both the environment and the offline datasets, keeping in tune with the online learning progression of the agent. Intuitively, by leveraging feedback from the online exploration of the agent, we can evaluate the effect of various mixing ratios for offline data replay, enabling a non-trivial switch that bypasses the need for extensive, context-specific empirical adjustments. Specifically, we develop an adaptive offline data replay mechanism by deploying a bandit-based model to explore and exploit the impact of different mixing ratios in the online learning process of the agent. This approach effectively determines the best configurations for mixing ratios. Furthermore, by strategically balancing the exploration and exploitation within the bandit model, we could enhance the effect of offline data replay while reducing excessive empirical adjustments to the pattern of mixing ratios. Our proposed mechanism can be incorporated into prevalent offline-to-online RL methods, illustrated by integrating it with two state-of-the-art and commonly used offline-to-online RL algorithms, specifically due to their performance and wide adoption (Kumar et al., 2020; Nakamoto et al., 2023). Our main contributions are as follows:

- We highlight the inherent challenges of existing offline data replay strategies in the online phase of offline-to-online RL, particularly the limitations of pre-determined and static mixing ratios. We emphasize the necessity for more adaptable ratio configurations, which could further harness the advantages of offline data without intensive empirical tuning.

- We propose an adaptive offline data mixing approach that dynamically adjusts mixing ratios of offline data replay with online feedback from the agent, enabling agents to effectively leverage offline data without context-specific adjustments.

- Comprehensive experiments validate the superiority of our proposed approach over state-of-the-art methods and its ability to identify adaptive offline data replay patterns across various environments and offline datasets.

## 2 RELATED WORK

**Offline-to-Online RL** A longstanding observation in reinforcement learning asserts that significant sampling is often required to achieve satisfactory performance from scratch (Ladosz et al., 2022; Moerland et al., 2023). In practical settings, a wealth of offline data is usually available, catalyzing the main premise behind offline-to-online RL (Levine et al., 2020). The fundamental paradigm of offline-to-online RL capitalizes on initializing a robust policy and value function pre-trained from offline data, segueing into subsequent online fine-tuning (Zheng et al., 2023; Nakamoto et al., 2023). This transition predominantly employs offline RL strategies grounded on policy constraints or pessimism, perpetuating training with congruent techniques in the online phase (Kostrikov et al., 2021; Kumar et al., 2020). Both theoretical and empirical results have highlighted the efficacy of such strategies (Xie et al., 2021; Song et al., 2022; Wagenmaker & Pacchiano, 2023). However, the performance of the learned policy can be constrained due to the inherent limitations of a static dataset, potentially hindering exploration during the online phase (Nair et al., 2020; Kostrikov et al., 2021; Ball et al., 2023). The challenge of maximizing the learning potential from both offline data and online interactions remains an active research topic. Several strategies have been investigated to finetune an offline-trained policy online, encompassing techniques such as parameter transferring (Rajeswaran et al., 2017; Xie et al., 2021), policy regularization (Rudner et al., 2021), and balancing between offline-online replay data during the online phase (Lee et al., 2022; Zhang et al., 2023). Among these strategies, the last approach could be seamlessly deployed with current offline-to-online RL algorithms, and has achieved excellent results in multiple applications (Nakamoto et al., 2023; Zheng et al., 2023).

**Offline Data Replay for Offline-to-Online RL** Efficiently leveraging offline data during the online training phase in offline-to-online reinforcement learning to augment the learning trajectories of the agent has garnered significant attention in recent reinforcement learning research. Earlier works primarily focused on harnessing high-quality demonstration data to facilitate the initial learning phase of agents, especially in complex scenarios with visual inputs (Rajeswaran et al., 2017; Nair et al., 2018; Rudner et al., 2021; Hansen et al., 2022). Recent advancements have underscored

that with well-crafted algorithm designs, the incorporation of offline data into the online phase can straightforwardly and effectively augment agent learning (Ball et al., 2023; Nakamoto et al., 2023). Notably, these recent methods, which seamlessly integrate offline data without imposing explicit constraints, have yielded promising results. The predominant strategies primarily consist of initializing the replay buffer with offline data (Vecerik et al., 2017; Hester et al., 2018), or adopting a balanced sampling strategy to manage both online and offline data sources (Kalashnikov et al., 2018; Hansen et al., 2022; Lee et al., 2022; Zhang et al., 2023; Nakamoto et al., 2023; Zheng et al., 2023). In our empirical investigations, we too affirm that appropriate offline data utilization can significantly enhance the online reinforcement learning phase. However, we also observe that a blanket application of a singular strategy across a vast spectrum of environments and datasets often falls short. This inadequacy is prominently evident as varying environments and the nature of offline data often dictate distinct optimal choices for offline data replay.

## 3   DATA UTILIZATION IN OFFLINE-TO-ONLINE REINFORCEMENT LEARNING

**Data Connections Between Offline RL and Online RL**   Offline and online RL are inherently intertwined in many ways. Offline RL offers sample efficiency, operating without the need for online interactions. However, its performance is often limited by the static nature of its offline dataset. On the other hand, online RL, although providing potentially higher performance capabilities, is not as sample-efficient. To harness the combined strengths of both paradigms, the field of offline-to-online RL aims to forge a seamless transition from the offline training to online training, and a promising avenue within this goal is optimizing the utilization of offline data during online training. Formally speaking, for a given value network $Q_\phi$ and a policy network $\pi_\theta$, an offline RL algorithm will first be deployed to pre-train the agent with an offline dataset $\mathcal{D}_{\text{offline}}$. Subsequent training then occurs in an online environment, where the agent interacts and learns from its online replay buffer $\mathcal{D}_{\text{online}}$. However, as noted in previous work, it is exceptionally difficult to first train a policy using offline data and then direct improve it using online RL due to distribution shift (Nair et al., 2020), and a simple yet efficient approach to resolve this problem is to incorporate the offline data in the online training phase (Lee et al., 2022; Ball et al., 2023; Nakamoto et al., 2023). One common strategy is to populate the online replay buffer with offline data, enabling the agent to learn from the union of the offline data and the online data $\mathcal{D}_{\text{offline}} \cup \mathcal{D}_{\text{online}}$. Another approach is to employ a mixing ratio, facilitating balanced sampling from both data sources. Specifically, given a mixing ratio $m \in [0, 1]$ representing what percentage of offline and online data is seen in each batch during online training, for each batch we sample $m \times 100\%$ from the offline data and $(1-m) \times 100\%$ from the online data.

**Towards Effective Replay of Offline Data**   While leveraging a pre-determined mixing ratio to utilize offline data during online training has proven effective, there is not a one-size-fits-all optimal choice for every environment or offline dataset. Historically, many have adopted a 50% mixing ratio, sometimes termed as "symmetric sampling" (Ball et al., 2023; Nakamoto et al., 2023). However, as an example shown in Figure 1, in the "mixed" task, a symmetric sampling could effectively improve the online training performance, while in the "complete" task, a lower mixing ratio performs much better. Identify an effective pattern of mixing ratio for some environment often requires experimental, context-specific adjustments. Intuitively, the complexity of the environment and the quality of the offline dataset could heavily influence the optimal mixing ratio for offline data replay. For example, for a tough environment with high observation dimensions or sparse rewards, agents might find early online exploration challenging. Here, starting the online phase with a higher mixing ratio could help retain lessons from offline demonstrations. But a high mixing ratio might also be harmful to the efficiency of online explorations, since the agent gets less chance to learn from its online explored data. The nature of

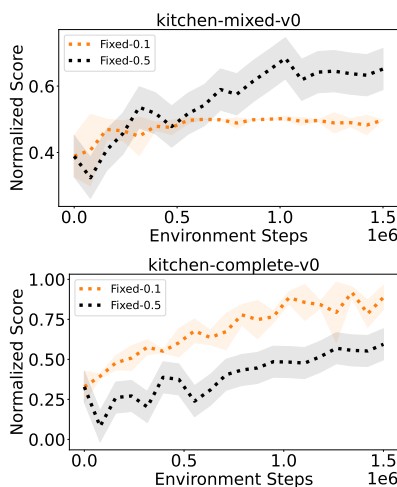

Figure 1: The effect of different mixing ratios on two FrankaKitchen tasks.

plorations, since the agent gets less chance to learn from its online explored data. The nature of

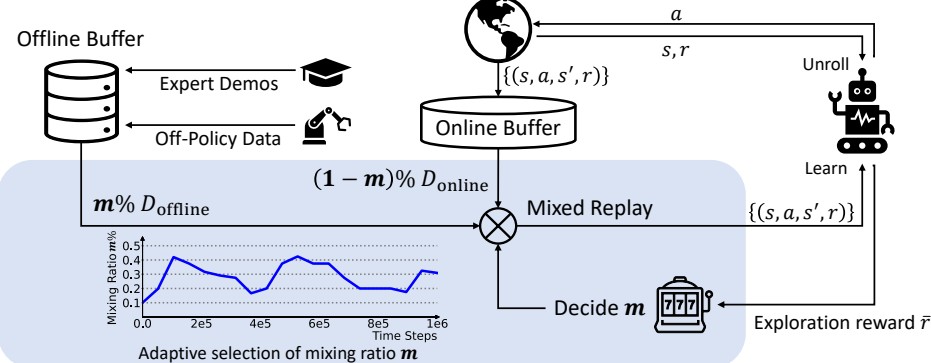

Figure 2: **Illustration of the online training scheme of ROAD.** In the online training phase, ROAD controls the offline data replay pattern by selecting the mixing ratio in an adaptive fashion with the training progression to facilitate online training. The mixing ratio $m$ adjusted by ROAD could control the mixed replay for the offline dataset and the online buffer.

the offline dataset also weighs in. High-quality datasets, such as those with expert demonstrations, might justify a higher mixing ratio to allow the agent to exploit this superior knowledge. Conversely, if the dataset mainly consists of medium or low quality transitions, a reduced mixing ratio during online training might be more beneficial. Moreover, the best mixing ratio may vary as the learning progresses of the agent. In the early stages of online learning, a higher mixing ratio can help the agent learn from offline demonstrations and maintain its offline-trained capabilities. As the agent refines its online policy over time, reducing the mixing ratio might enhance policy exploitation. In essence, the optimal mixing ratio is context-dependent, varying across environments, offline datasets, and stages of the learning journey of the agent.

## 4 ADAPTIVE OFFLINE DATA REPLAY FOR OFFLINE-TO-ONLINE RL

In this section, we present an adaptive offline-data replay scheme for the online training phase of offline-to-online RL. Importantly, our proposal is not bound to any specific offline-to-online RL algorithms and can seamlessly integrate with various value-based offline-to-online methods. The efficacy of this combined approach, however, may vary based on the specific methods selected.

### 4.1 OPTIMIZING OFFLINE DATA REPLAY

Addressing the challenges highlighted earlier, we introduce a novel offline data replay approach, compatible with existing offline-to-online RL algorithms, termed Reinforcement Learning with Optimized Adaptive Data-mixing (ROAD). This method is visualized in Figure 2. At its core, we first adhere to the standard offline-to-online RL process, pre-training on an offline dataset to derive a policy $\pi_\theta$ and a Q-function $Q_\phi$. To enhance the use of both offline and online data during the online phase, unlike previous methods that initialized the online buffer with offline data or used a pre-determined mixing ratio, ROAD maintains a optimistic value estimate for various mixing ratios. It then dynamically adjusts the chosen mixing ratio by consistently exploring and exploiting its value estimation.

To elaborate, given a set of potential mixing ratios $\mathcal{M}$, ROAD upholds a value estimate $R \in \mathbb{R}^{|\mathcal{M}|}$, gauging the effectiveness of different mixing ratios in promoting online learning. Furthermore, ROAD employs a bandit-based approach to strike a dynamic balance between exploring and exploiting different mixing ratios throughout the training of the agent. This is achieved by maintaining an upper confidence bound (UCB) value for different mixing ratios, guiding their exploration and selection (Lattimore & Szepesvári, 2020). Specifically, between two mixing ratio update intervals, ROAD captures the average reward of the agent from online explorations. If the agent has sampled $n$ trajectories $\{\text{traj}_1, \ldots, \text{traj}_n\}$ within this interval, the average exploration reward can be

---

**Algorithm 1** ROAD: Reinforcement Learning with Optimized Adaptive Data Mixing

---

**Require:** offline pre-trained policy $\pi_\theta$ and Q function $Q_\phi$, online RL algorithm $\{L_{\text{online}}^{Q_\phi}, L_{\text{online}}^{\pi_\theta}\}$, learning rates $\{\lambda_Q, \lambda_\pi\}$, candidate set of mixing ratios $\mathcal{M}$, hyper-parameter $c$ for UCB

1: **Initialize:** offline replay buffer $\mathcal{D}_{\text{offline}}$ with offline data, online replay buffer $\mathcal{D}_{\text{online}} = \emptyset$
2: ROAD setup: $R_i \leftarrow 0$, $N_i \leftarrow 0$ for each arm $i$ in $\mathcal{M}$; initialize $m$ as a random element of $\mathcal{M}$
3: **while** in online training phase **do**
4:     **for** each environment step **do**
5:         $a_t \sim \pi_\theta(a_t|s_t)$, $s_{t+1} \sim T(s_{t+1}|s_t, a_t)$, $\mathcal{D}_{\text{online}} \leftarrow \mathcal{D}_{\text{online}} \cup \{(s_t, a_t, r(s_t, a_t), s_{t+1})\}$
6:     **end for**
7:     Update $R_m$ according to Eq. (2), $N_m \leftarrow N_m + 1$
8:     Calculate the UCB values and choose mixing ratio $m \in \mathcal{M}$ according to Eq. (3)
9:     **for** each gradient step **do**
10:         % online training using batches from $m \times 100\% \times \mathcal{D}_{\text{offline}}$ and $(1-m) \times 100\% \times \mathcal{D}_{\text{online}}$
11:         $\phi \leftarrow \phi - \lambda_Q \nabla_\phi L_{\text{online}}^{Q_\phi}(\phi), \quad \theta \leftarrow \theta - \lambda_\pi \nabla_\theta L_{\text{online}}^{\pi_\theta}(\theta)$
12:     **end for**
13: **end while**

---

computed as

$$\bar{r} = \frac{1}{n} \sum_{i=1}^{n} \text{Reward}(\text{traj}_i), \tag{1}$$

where $\text{Reward}(\cdot)$ is a function determined by the environment that calculates the total reward of one trajectory. The average exploration reward $\bar{r}$ is then utilized by ROAD in a cumulative moving average way to update the unbiased value estimate of the previously selected mixing ratio $m$

$$R[m] = \frac{\bar{r} - R[m]}{N[m]}, \tag{2}$$

where $R[m]$ is the value estimate of $m$, and $N[m]$ is the times $m$ has been selected by ROAD before. Subsequently, ROAD updates the mixing ratio $m$ with the highest UCB value for the agent

$$m = \arg\max_{m'} \left\{ R[m'] + \sqrt{\frac{c \cdot \text{sum}(N)}{N[m']}} \right\}, \tag{3}$$

where $c$ is the exploration parameter for UCB values, and and $\text{sum}(N)$ is the total number of times different arms have been pulled. With the current mixing ratio $m$ in play, batches are randomly sampled by $m \times 100\%$ from the offline dataset and $(1-m) \times 100\%$ from the online replay buffer, for the online learning algorithm to conduct gradient updates.

## 4.2 OFFLINE-TO-ONLINE RL WITH ROAD

We focus on adaptive offline data replay in offline-to-online reinforcement learning, and the dynamic mixing ratio adjustment approach provided by ROAD could be integrated with conventional value-based offline-to-online reinforcement learning algorithms. Concretely speaking, given an offline dataset and an online environment, the parameterized policy function $\pi_\theta$ and Q function $Q_\phi$ will first be pre-trained with the offline dataset, then the agent will enter the online training phase, where ROAD, or other offline data replay approaches, will replay the offline data along with the online-collected data for the online RL algorithm to learn. Specifically, ROAD will also update its value estimations in the online training steps of the agent, and the entire procedure is summarized in Algorithm 1, where the data collection and gradient update processes follows those in algorithms like Cal-QL (Nakamoto et al., 2023) and CQL (Kumar et al., 2020), and we initialize the offline replay buffer $\mathcal{D}_{\text{offline}}$ using existing data and expand the online replay buffer with new transitions during the online phase (line 5). After executing multiple environment steps, the average exploration reward is calculated, and the UCB estimates within ROAD are updated with the recent transitions (line 7-8). ROAD then determines the mixing ratio for the upcoming iteration. The gradient updates follow, acting on a combined replay of $\mathcal{D}_{\text{offline}}$ and $\mathcal{D}_{\text{online}}$, with batches determined by the policy of ROAD (line 10-11).

## 5 EXPERIMENT

In this section, we first evaluate the effectiveness of ROAD on various types of benchmark tasks with comparison to a number of baseline methods. Then we further show an analysis and discussion about the selection behavior of ROAD. Besides, We conduct experiments about the parameter sensitivity of ROAD and its performance across different dataset qualities.

### 5.1 OFFLINE-TO-ONLINE RL EXPERIMENTS

**Tasks and Settings** The goal of our experimental evaluation is to study how well ROAD can facilitate efficient training in the online phase. To this end, we evaluate ROAD across several benchmark tasks: **(1)** AntMaze tasks from D4RL (Fu et al., 2020): This task involves controlling an ant quadruped robot to navigate from a start point to a predetermined goal location within a maze. An agent is rewarded with a score of $+1$ upon reaching a pre-designated radius surrounding the goal; otherwise, it receives a score of $0$. **(2)** Adroit dexterous manipulation tasks Ashvin et al. (2020); Kostrikov et al. (2021): These tasks focus on mastering intricate manipulation skills using a 28-DoF five-fingered hand. The tasks entail manipulating a pen within the hand to achieve a desired orientation (pen-binary) or relocating a ball to a set location (relocate-binary). **(3)** FrankaKitchen tasks from D4RL: This task necessitates controlling a 9-DoF Franka robot to reach a target configuration within a kitchen setup. A successful policy mandates completing four sub-tasks within a single rollout in the kitchen environment, earning a binary reward of $+1$ or $0$ for each completed sub-task. The offline datasets for the D4RL tasks are provided by the open-source benchmark. In contrast, the Adroit tasks utilize a limited offline dataset comprising 25 demonstrations sourced from human teleoperation and additional trajectories gathered by a behavior cloning policy (Nakamoto et al., 2023). For the offline-to-online reinforcement learning algorithms, we choose two representative algorithms that enable fast online finetuning: Conservative Q-learning (Kumar et al., 2020) and Calibrated Q-learning (Nakamoto et al., 2023), to test ROAD and the baselines. The candidate set $\mathcal{M}$ is set to be $\{10\%, 20\%, 30\%, 40\%, 50\%\}$, and the exploration parameter $c$ is set to be $1.0$. Each training is repeated with four different random seeds. The implementation details and hyper-parameters for Cal-QL and CQL are listed in Appendix C.

**Baselines** We compare ROAD with the following baselines that utilize offline data in the online training phase in various ways: **(i) `Buffer`** (Vecerik et al., 2017; Hester et al., 2018): initialize the online replay buffer $\mathcal{D}_{\text{online}}$ as the offline buffer $\mathcal{D}_{\text{offline}}$, then add new online samples afterward. **(ii) `Fixed`** (Zhang et al., 2023; Nakamoto et al., 2023; Zheng et al., 2023): conducting the online training phase on a mixture of the offline data and the new online data, weighted in some fixed mixing ratio during fine-tuning. It has been reported that symmetric sampling (mixing ratio = $50\%$) is effective across a variety of scenarios (Ball et al., 2023; Zheng et al., 2023), and here we will compare ROAD with fixed mixing ratio in $\{0\%, 10\%, 20\%, 30\%, 40\%, 50\%\}$. **(iii) `Decreasing`**: decreasing the mixing ratio gradually is a standard baseline for dynamic adjustments of the mixing ratio to mitigate the distribution shift from the offline dataset to the online environment, and here we will decrease the mixing ratio linearly from $50\%$ linearly to $0\%$, but keep the minimum value at $10\%$ for better performance: $m = \max\left\{0.5 \times \left(1 - \frac{\text{env\_step}}{\text{max\_env\_step}}\right), 0.1\right\}$. **(iv) `Uniform`**: conduct uniform selection of the mixing ratios among the arms same as ROAD.

**Performance Analysis of Various Offline Data Replay Methods** The normalized return curves across all tasks can be seen in Figure 3 for Cal-QL and Figure 4 for CQL. A general improvement in performance is observed after online training compared to the initial offline pre-training outcomes. Pure online training, denoted as **`Fixed-0.0`**, consistently lagged in performance and displayed the highest variance across tasks. For specific environments, such as the Adroit tasks with Cal-QL or the AntMaze tasks with CQL, the **`Buffer`** strategy demonstrated better performance than **`Fixed-0.0`**, indicating the benefits of integrating offline data during online buffer initialization. However, in some other environments, the **`Buffer`** approach either slowed down the learning process or led to decreased performance. The effectiveness of **`Buffer`** seems to depend on factors such as task complexity, the quantity and quality of offline datasets, and the backbone algorithm chosen, echoing findings from prior studies (Ball et al., 2023; Zhang et al., 2023). The results of the **`Fixed`** method varied based on the mixing ratio selected. Some environments yielded similar results across different fixed mixing ratios, while others showed pronounced differences. Identifying the optimal

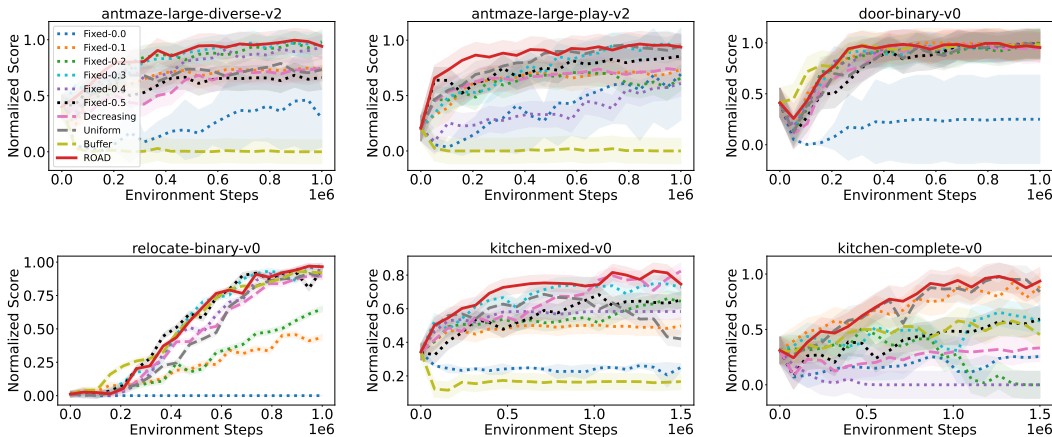

Figure 3: Normalized return curves of different offline-data replay methods on the benchmark tasks, using Cal-QL (Nakamoto et al., 2023) as the backbone.

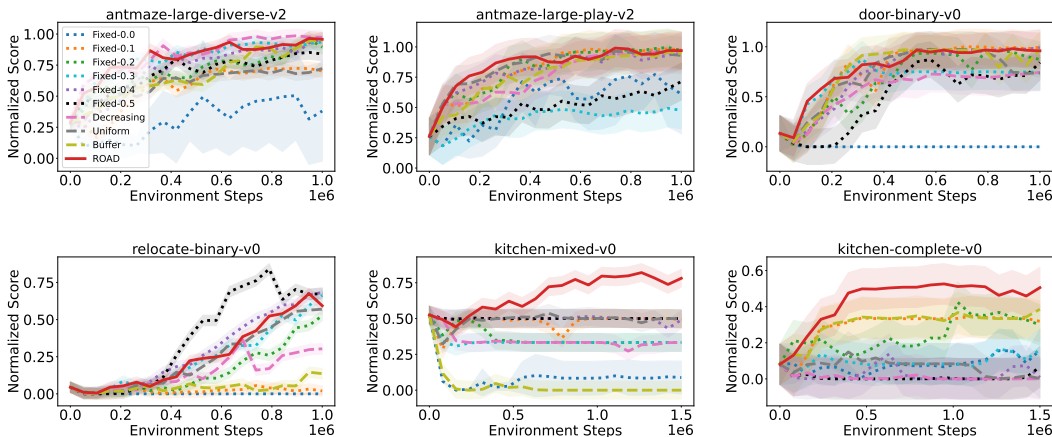

Figure 4: Normalized return curves of different offline-data replay methods on the benchmark tasks, using CQL (Kumar et al., 2020) as the backbone.

fixed mixing ratio often required empirical tuning specific to each environment and dataset. More-over, sticking to a fixed ratio during online learning may not always be the most effective. This is evident with the **Decrease** strategy, which surpassed some fixed mixing ratios in some environments by progressively reducing the mixing ratio. However, its effectiveness is not consistent across all settings. The **Uniform** approach also produced mixed results: in certain scenarios, it could match or outperform the performance of **Fixed**, while in others it leads to worse performance than the best fixed mixing ratio.

**Performance of ROAD**    In contrast to the previously discussed methods, ROAD consistently performed at a level that either matched or surpassed most baselines across a wide range of environments and algorithmic backbones. Compared to various **Fix** configurations, ROAD consistently aligned with or outperformed the optimal fixed mixing ratio. Furthermore, when juxtaposed with both **Decrease** and **Uniform**, the performance of ROAD remained the most stable and consistent. This proves its ability to adapt across diverse environments and algorithms, reinforcing its robustness and adaptability in offline-to-online RL scenarios.

## 5.2 ADAPTIVE MIXING RATIO SELECTION OF ROAD

A distinguishing feature of ROAD, in contrast to prior works that adopt a pre-determined mixing ratio, is its bandit-based model which allows ROAD to make dynamic decisions on the subsequent

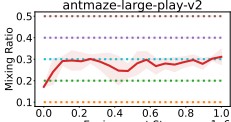 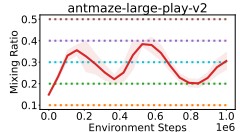 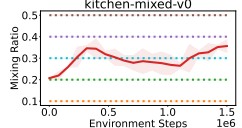 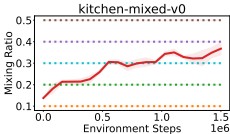

(a) Mixing ratio pattern generated by ROAD in AntMaze play task under Cal-QL (left) and CQL (right).

(b) Mixing ratio pattern generated by ROAD in FrankaKitchen mixed task under Cal-QL (left) and CQL (right).

Figure 5: Visualization of mixing ratio pattern generated by ROAD in different environments.

mixing ratios, based on its estimation of the value of various mixing ratios. This inherently renders its offline replay pattern more adaptive. Notably, our experiments have showcased that the efficacy of fixed mixing ratios varies significantly across different environments. This drives our exploration into how ROAD adapts its mixing ratio choices in light of the underlying task and dataset conditions. To illustrate this, we traced the evolution of the mixing ratio choices of ROAD over online training steps in two challenging environments: AntMaze play and FrankaKitchen mixed, as depicted in Figure 5. Our primary observations are:

1. *Adaptive Ratio Selection:* Across the spectrum of available mixing ratios, ROAD aptly identifies and converges towards relatively appropriate mixing ratio patterns. For instance, when integrated with the Cal-QL algorithm, ROAD's mixing ratio decisions in both tasks gravitate towards approximately 0.3. Significantly, a fixed 0.3 mixing ratio also yielded near-optimal results for these environments under the Cal-QL setup. Besides, with its explorations of various mixing ratio selections, ROAD achieves superior results during the early-to-mid online learning phase than the fixed 0.3 ratio.

2. *Diverse Patterns across Algorithms:* For the same environment, ROAD exhibits adaptability in its mixing ratio pattern based on the algorithm it is integrated with. Taking the AntMaze play task with the CQL algorithm as an instance, ROAD, when combined with CQL, learns a fluctuating mixing ratio pattern that oscillates between the 0.2 to 0.4 range. As our results indicate, this range aligns with the fixed mixing ratios where CQL performs most optimally for this task. For the FrankaKitchen mixed task, the strategy of ROAD with CQL gravitates towards incrementally increasing the mixing ratio. This distinct pattern becomes evident when we contrast the performance of agents trained with a static mixing ratio against those guided by ROAD. The latter consistently outperforms the former, indicating that an incremental mixing ratio during online training can efficiently leverage offline data for enhanced performance.

### 5.3 PARAMETER SENSITIVITY STUDIES

One of the essential components in the design of ROAD is the utilization of an upper confidence bound (UCB) to balance exploration and exploitation of different mixing ratios. A pivotal factor in this design is the weight parameter $c$ assigned to the exploration term in UCB computation. A low weight often prompts the algorithm to converge after some exploration, while a higher weight encourages more extensive exploration across mixing ratios. Given the significant performance disparities attributed to different mixing ratios across varying algorithms and environments, it becomes crucial to gauge its sensitivity to this exploration weight.

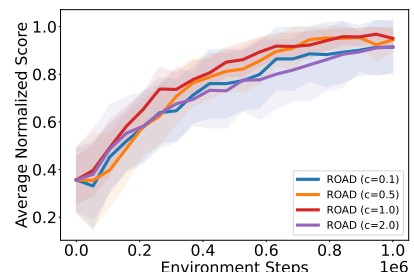

Figure 6: Average normalized score with various exploration parameters.

We evaluated the average performance of ROAD across several tasks in both AntMaze and Adroit, under different exploration parameters. The average result for selected parameters are depicted in Figure 6. As can be discerned from the figure, the choice of exploration parameter does influence the overall performance. Specifically, opting for a smaller exploration pa-

rameter (e.g., 0.1) tends to undercut the exploration of ROAD across mixing ratios, leading to a mild degradation in overall performance. On the other hand, a larger exploration parameter (e.g., 2.0) can overemphasize exploration, posing challenges in convergence and subsequently diminishing the results. However, on the whole, while our chosen $c = 1.0$ yielded relatively better outcomes, the overall performance of ROAD remains commendably stable across the different parameter selections. This stability underscores the robustness of ROAD in learning adaptive offline data replay. Some extensive examination of the UCB strategy deployed in ROAD is discussed in Appendix B.

### 5.4 Performance Across Dataset Qualities

We further analyze the adaptability of ROAD across diverse environments and offline datasets. Previous experiments have demonstrated its capacity to adapt across different datasets in similar environments. For instance, within AntMaze tasks, the "diverse" dataset contains trajectories to random goals, whereas the "play" dataset has trajectories to hand-picked locations. Although different algorithm and mixing ratio combinations showed varied results across these datasets, the adaptive selection of ROAD consistently yielded strong performances. A similar trend was observed in FrankaKitchen tasks. While the "complete" dataset contains demonstrations of all four target subtasks completed in sequence, the "mixed" dataset only includes random sub-tasks without the four primary ones in order. Here, ROAD also stood out in terms of performance across algorithms.

Given the relatively high quality of standard benchmark datasets, we conducted an additional test to assess the capability of ROAD on lower-quality datasets, mirroring scenarios like autonomous driving where offline data might be of average quality. For this, we purposefully degraded dataset quality by retaining only the lower reward samples. Taking the relocate-binary environment as an example, ROAD, along with fixed mixing ratios of $0.3$ and $0.5$, all performed well under the Cal-QL algorithm. However, when we further reduced the dataset quality by keeping only zero-reward samples, all algorithms showed diminished performance, as shown in Figure 7. Despite this reduction, ROAD still showcased superior learning outcomes compared to other offline data replay approaches, indicating its resilience and adaptability to varying dataset qualities.

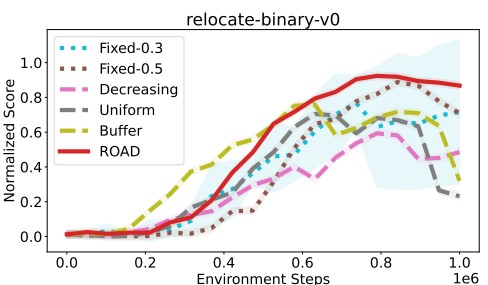

Figure 7: Performance of ROAD with Cal-QL under a low-quality offline dataset.

## 6 Conclusions, Limitations and Future Work

In this work, we emphasized the importance of designing efficient offline data replay in the online training of offline-to-online RL. Our goal was to leverage the benefits of both offline and online methods without extensive context-specific adjustments. We introduced a mixing ratio selection scheme as a stride in this direction. This method is straightforward and works well with existing offline-to-online RL algorithms, which we showed using two different combinations. The effectiveness of our method was proven across multiple benchmark tasks and algorithm backbones.

While achieving promising performance, the proposed approach also has some limitations. Currently, we only explore and utilize different mixing ratios without considering other offline data replay techniques. While our results have been consistent and effective within this scope, there might be room for improvement by integrating more offline data replay methods into its selection pool, like initializing the online replay buffer with offline data. Another potential avenue of exploration lies integrating our method with importance sampling techniques. The distribution shift between offline data and samples from the online environment, coupled with the computational demands potentially posed by offline data, necessitates the crafting of more refined importance sampling methods. One proposition to address this challenge entails deploying a neural network to estimate the density ratio, but the efficacy of such approaches has not been observed consistently on a broad scale yet. We leave the exploration of similar combinations and generalization as an interesting future work.

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

## A    ENVIRONMENT DETAILS

### A.1    ANTMAZE TASKS

Antmaze tasks challenge an agent to guide an 8-DoF ant robot through a maze, starting from one point and ending at a set goal. A simple reward system is used: the agent gets a reward of $+1$ if the goal is reached and $0$ otherwise. We primarily focus on intricate mazes and incorporate two D4RL datasets: large-diverse and large-play. While the "diverse" dataset is filled with trajectories from arbitrary start points to random goals, the "play" dataset has trajectories directed at specific, non-goal locations. Each task ran for 1000 episodes. Both Cal-QL and CQL agents were initially trained with the offline datasets for 1M steps. Subsequent online fine-tuning was conducted over 1M environment steps, integrating ROAD or competing offline data replay strategies.

### A.2    ADROIT TASKS

The Adroit arena demands control over a sophisticated 24-DoF shadow hand robot. The chosen tasks for this domain were relocate-binary (listed twice, assuming a typo). These tasks revolve around a limited set of expert human data (roughly 25 sets), bolstered by trajectories from a behavior-cloned policy. Given the tight dataset distribution and expansive action space, the Adroit tasks are inherently challenging due to sparse rewards and exploration issues. During the offline training phase, agents underwent training for 20 K steps. Online fine-tuning spanned 1M environment steps for both tasks. Both tasks had a 200-episode duration.

### A.3    FRANKA KITCHEN TASKS

The tasks within the Franka Kitchen domain task an agent with manipulating a 9-DoF Franka robot to set up a kitchen based on a specific layout. The task breaks down into 4 distinct subtasks. The agent earns rewards on a scale from $0$ to $+4$, based on the number of subtasks successfully executed. Completing the overall task demands mastering individual subtasks and deducing the right sequence. We evaluated agents on this domain utilizing two contrasting datasets: kitchen-complete and kitchen-mixed. The "complete" dataset encompasses full task sequences, while the "mixed" dataset presents fragmented sequences without a complete demonstration, pushing agents' generalization abilities to the limit. Each task here also had a length of 1000 episodes. Initial training using the offline dataset spanned 500 K steps for both Cal-QL and CQL. Following this, online fine-tuning was executed across 1.25M environment steps, deploying either ROAD or other alternative offline replay strategies.

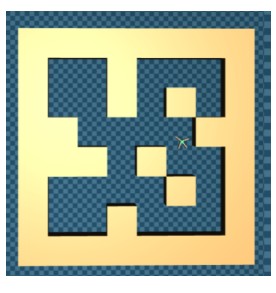
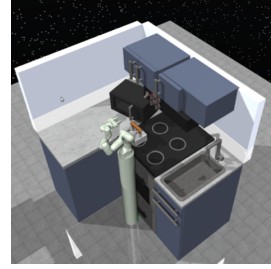

(a) AntMaze                (b) Adroit                (c) Franka Kitchen

Figure 8: Benchmark environments of ROAD.

## B    DELVING DEEPER INTO BANDITS IN ROAD

In this section, we dissect the rationale behind employing the UCB strategy within the ROAD framework. A central concern here is the latency inherent in the average exploration reward garnered during an agent's online exploration. Such latency suggests that the effect of a specific mixing ratio might be manifested after a certain lag. This opens the door to other adaptive selection strategies, besides UCB (Burtini et al., 2015). To critically assess UCB's adaptability across varied environmental

feedbacks, we juxtaposed the UCB-centric ROAD implementation against a few other potential configurations. Two main axes of exploration emerge:

1. *Bandit with Nonstationary Feedback:* This direction is inspired by the potential volatility in the agent's online exploration reward, a characteristic especially pronounced in the early stages of training. Furthermore, a nonstationary design captures the dynamism in the efficacy of various mixing ratios across different stages of agent training. We particularly honed in on two distinct implementations:

   - **DiscountedUCB (Garivier & Moulines, 2008):** This algorithm factors in a discount when computing value estimates based on past rewards. It emphasizes the significance of more recent rewards over the distant ones.
   - **SlidingWindowUCB (Garivier & Moulines, 2011):** Here, the value of different mixing ratios is estimated within a recent time window, thereby focusing on more immediate rewards. The primary goal of these adaptations is to discern whether UCB can adapt to potential fluctuations in exploration rewards during online exploration.

2. *Bandit with Delayed Feedback:* The core premise here is the lagged effect that a particular mixing ratio might induce on the agent's online learning. This means a set mixing ratio might manifest its impact only after a certain period. Taking this into account, we present:

   - **DelayedUCB (Grover et al., 2018; Vernade et al., 2020):** This approach remodels UCB by accounting for the possible delayed feedback. Such feedback is represented as a process compliant with a geometric distribution. We've substantiated the convergence properties of DelayedUCB in a theoretical setting.

By substituting the UCB estimation of the mixing ratio in ROAD with the aforementioned methods, we aspire to ascertain if the role of UCB-based estimation remains consistent and robust within the ROAD framework. We conduct experiments on Antmaze-large-diverse, Antmaze-large-play, Adroit-door-binary and Adroit-relocate-binary, and the result with Cal-QL is shown in Figure 9. It could be seen that these algorithms keep about the same performance among these tasks, showcasing the adaptivity of UCB in ROAD for mixing ratio exploration and exploitation, which is shown to be simple yet efficient in multiple environments and tasks.

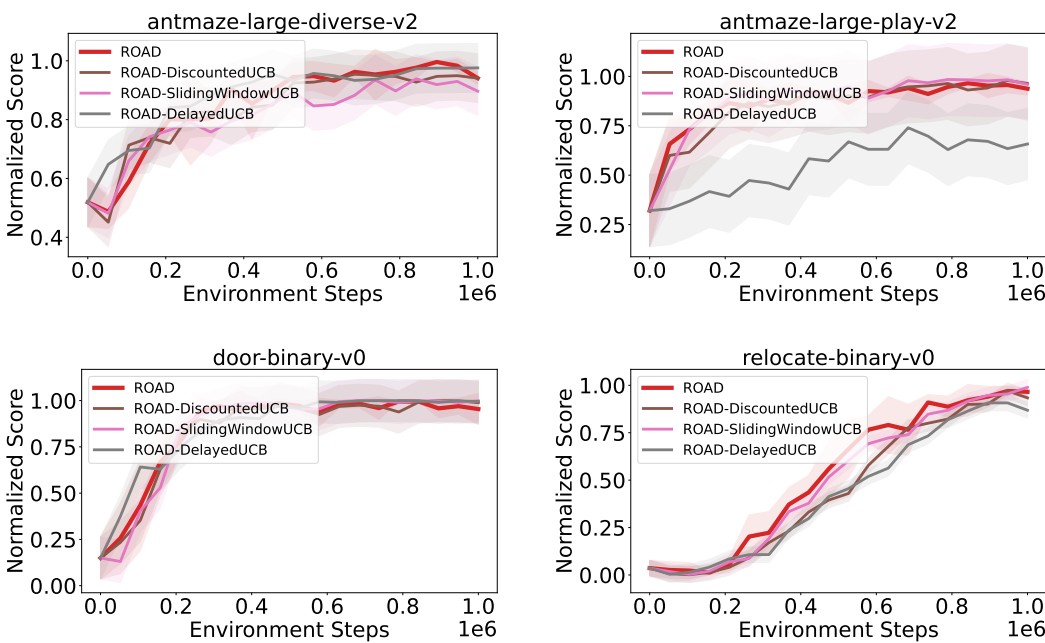

Figure 9: Normalized return curves of ROAD and other ROAD-based offline-data utilization methods but replace UCB with DiscountedUCB/SlidingWindowUCB/DelayedUCB.

## C   EXPERIMENT DETAILS

### C.1   NORMALIZED SCORE CALCULATION

Since the AntMaze and Adroit domains are goal-oriented, sparse reward tasks, we computed the normalized metric as simply the goal achieved rate for each method in these domains. For the kitchen tasks, each task is to solve a series of 4 sub-tasks that need to be solved in an iterative manner, so the normalized score is computed as the proportion of the sub-tasks that have been solved in all tasks.

### C.2   HYPER-PARAMETERS

We list the hyperparameters in ROAD with Cal-QL and CQL in Table 1. The code will be made publicly available upon acceptance of this paper.

Table 1: Main hyper-parameters in ROAD.

| Hyper-parameter | Value |
|---|---|
| batch size | 256 |
| replay buffer size | 1000000 |
| discount | 0.99 |
| policy learning rate | 1e-4 |
| Q function learning rate | 3e-4 |
| orthogonal init | True |
| policy arch | 256-256 |
| Q function arch | 256-256-256-256 |
| CQL min Q weight | 5.0 |
| CQL target action gap | 0.8 |

## D   EXPERIMENT ON A MORE VARIETY OF TASKS

In this section, we delve deeper into the performance of ROAD under varying task difficulties, comparing it against different baselines to analyze its effectiveness across diverse task complexities and the potential impact of adaptive offline data replay in these scenarios. Additionally, we contrast the average performance of a fixed mixing ratio with that of ROAD, assessing the potential improvements ROAD offers over the optimal mixing ratio determined under average conditions.

### D.1   COMPARISON WITH VARIOUS TASK DIFFICULTIES

Firstly, we focus on experiments tailored to different task difficulties. Here, we employ the widely-used D4RL benchmark, specifically selecting the "large" and "medium" variants within the AntMaze environment for testing. As depicted in Figure 10, the "large" environment, characterized by its larger maze size and complexity, poses a significantly higher task difficulty compared to the "medium" environment. For both "large" and "medium" environments, which include "diverse" and "play" datasets respectively, we further test using ROAD in conjunction with both the Cal-QL and CQL algorithms, with results illustrated in Figure 10 and Figure 11.

Our findings reveal a notable difference in the performance of various baseline algorithms across these different task difficulties. Specifically, in the "large" environment, the higher task complexity leads to significant performance disparities among different fixed mixing ratios. Moreover, the optimal fixed mixing ratio varies across datasets and offline-to-online RL algorithms. In contrast, in the "medium" environment, with relatively simpler task difficulty, the performance gap among various baseline algorithms is smaller. Notably, ROAD consistently achieves commendable performance in both "large" and "medium" environments, generally matching or exceeding the best fixed mixing ratio. It outperforms methods like buffer initialization and uniform selection of mixing ratios.

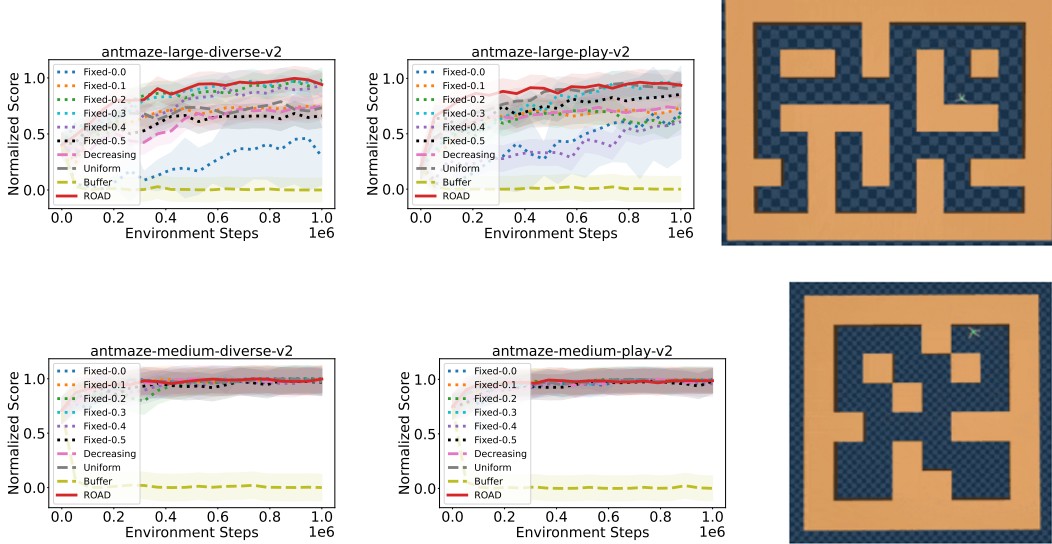

Figure 10: Normalized return curves of AntMaze large tasks (diverse and play) and AntMaze medium tasks (diverse and play), using Cal-QL (Nakamoto et al., 2023) as the backbone.

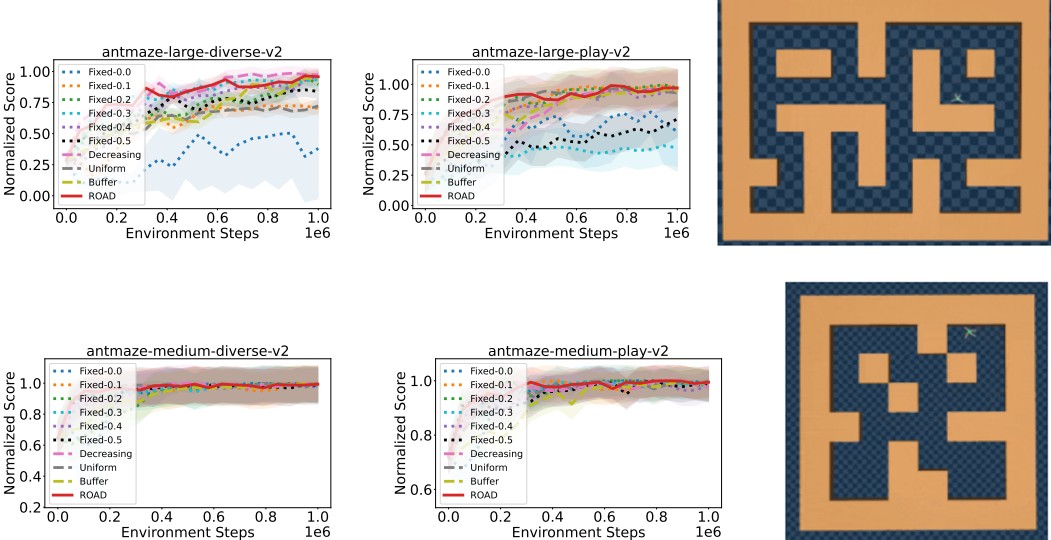

Figure 11: Normalized return curves of AntMaze large tasks (diverse and play) and AntMaze medium tasks (diverse and play), using CQL (Kumar et al., 2020) as the backbone.

This further underscores ROAD's adaptability across varying task complexities, demonstrating its capability to excel whether there's a high or low distinction among different mixing ratios.

Additionally, we extended our examination of ROAD in conjunction with various offline-to-online RL algorithms across a broader range of environments. While our initial experiments focused on the "door" and "relocate" environments within Adroit, we have now conducted further testing and verification in the "pen" environment. It is important to note that previous studies have highlighted the suboptimal performance of the CQL algorithm in the "pen" setting, whereas Cal-QL tends to achieve better results. Our experimental outcomes, as shown in Figure 11, involved trials with ROAD integrated with both Cal-QL and CQL, alongside comparisons with different baselines.

Our findings indicate that when combined with Cal-QL, ROAD is capable of matching the performance of the optimal baseline (in this case, a fixed 0.5 mixing ratio). However, when integrated

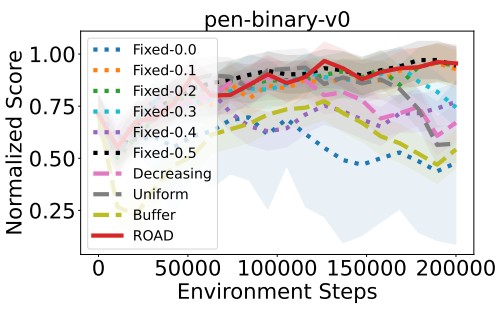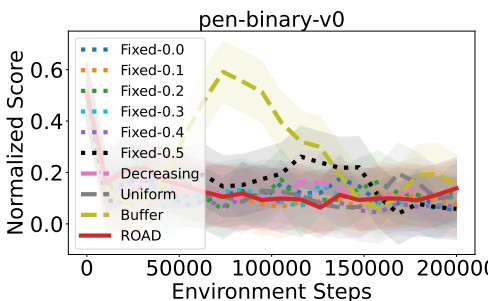

Figure 12: Normalized return curves of Adroit pen-binary-v0 environment, using Cal-QL (left) and CQL (right) as the backbone.

with CQL, ROAD's effectiveness is limited by the inherent constraints of the CQL algorithm, resulting in performance comparable to other baselines. It's worth mentioning that other offline data replay strategies, such as buffer initialization, also face limitations in this environment but seem to offer some performance improvements initially. This leads to an intriguing direction for future work: exploring how data replay methods can be synergistically integrated with the internal design of offline-to-online RL algorithms to further enhance performance during the online phase. This promising avenue of research could yield significant advancements in the field of reinforcement learning.

## D.2 AVERAGE PERFORMANCE OF FIXED MIXING RATIO AND ROAD

In this study, we evaluate the performance of different mixing ratios across multiple environments to determine the average optimal mixing ratio and compare it with the performance of ROAD. Section 5.3 of our paper assesses the impact of varying exploration parameters on ROAD within the Cal-QL framework. Here, we extend this analysis to include CQL, computing the average normalized return, as illustrated in Figure 14 (right). It is observed that while there are performance variations among different exploration parameters in CQL, the overall trend remains stable. Although an exploration parameter $c = 2$ shows the best overall performance, we consistently chose $c = 1$ as ROAD's exploration parameter in our experiments described in Section 5.1.

This finding suggests that further fine-tuning ROAD's parameters across different environments could yield even better overall performance. However, in this research, we opted for a uniform parameter selection to demonstrate ROAD's broad adaptability across various environmental difficulties and dataset qualities. This approach highlights ROAD's robustness and flexibility, underscoring its potential utility in a wide range of reinforcement learning applications.

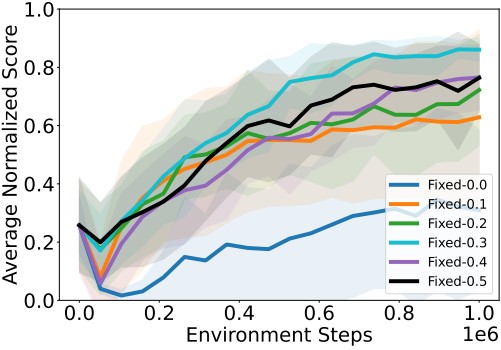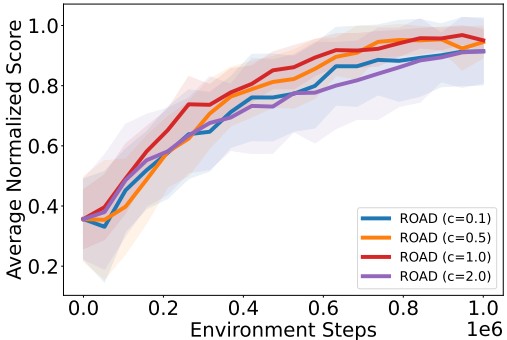

Figure 13: Average normalized score of fixed mixing ratios (left) and ROAD (right) with Cal-QL.

Furthermore, we conducted a similar analysis to assess the average performance of different fixed mixing ratios under both Cal-QL and CQL algorithms, with the results presented in Figure 13 and Figure 14. Notably, in our current Cal-QL experiments, a mixing ratio of **0.3** appears to yield the

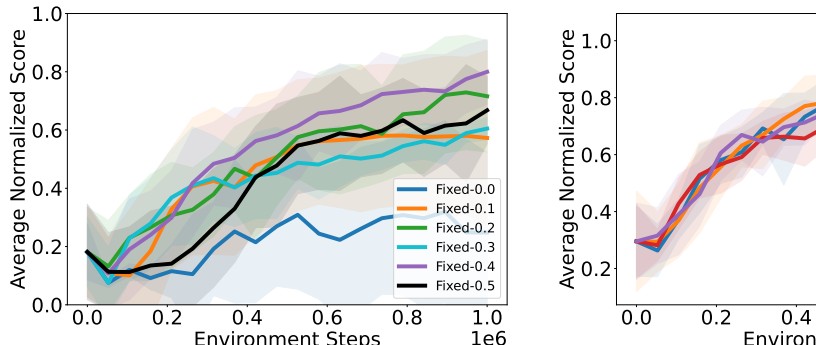

Figure 14: Average normalized score of fixed mixing ratios (left) and ROAD (right) with CQL.

best average performance, while in CQL experiments, a mixing ratio of **0.4** emerges as the most effective. This observation aligns with our earlier assertion that the choice of the optimal mixing ratio is influenced by the specific algorithm employed.

Upon comparing ROAD with the optimal fixed mixing ratio of $m^* = 0.3$ in Cal-QL and $m^* = 0.4$ in CQL, we found that the average performance of the optimal fixed mixing ratio is, in fact, inferior to that of ROAD with a chosen parameter ($c = 1$). Notably, in CQL, this is not even the average optimal mixing ratio for ROAD. This finding significantly demonstrates the flexibility and effectiveness of ROAD's adaptive offline data replay compared to a fixed mixing ratio. ROAD consistently achieves superior performance across various algorithms, environments of differing complexities, tasks, and qualities of offline datasets. This underscores the robustness and adaptability of ROAD in optimizing performance in a wide array of reinforcement learning scenarios.

## E    OPTIMALITY GUARANTEE OF ROAD

In this section, we present the optimality guarantee of the UCB algorithm as employed in the ROAD framework, specifically within the realm of multi-armed bandits. We demonstrate that if an optimal mixing ratio exists, ROAD is capable of achieving an $O(\log T)$ regret over $T$ rounds of interaction. The central premise revolves around the concept of a balanced replay in offline to online reinforcement learning, where a fixed mixing ratio is utilized during the online training phase.

**Assumption 1.** *Consider a set of $M$ arms, each representing a distinct mixing ratio. Among these, there exists an optimal arm, denoted as $m^*$, which yields the highest average reward. Each arm $i$ adheres to a reward distribution with an unknown mean $\mu_i$. The optimal arm $m^*$ is characterized by the highest mean reward, $\mu^* = \mu_{m^*}$.*

This assumption posits the existence of an optimal mixing ratio, analogous to the best strategy in previous offline to online RL algorithms, which is crucial for balanced replay.

**Assumption 2.** *The reward distribution for each arm is bounded. Specifically, for any given arm $i$, the reward $r_i$ at any time $t$ satisfies the constraint $0 \leq r_i(t) \leq 1$. This assumption typically aligns with environments featuring sparse rewards, where a trajectory is rewarded only upon task completion. Similar extensions can be considered for dense reward scenarios.*

**Theorem.**  *Within the context of these assumptions, the UCB algorithm implemented in ROAD achieves logarithmic cumulative regret over time. That is, the expected cumulative regret $R(T)$ after $T$ rounds of interaction is upper bounded as follows:*

$$R(T) \leq O(\log T). \tag{4}$$

**Proof.** Similar to Eq. (3), the UCB for arm $i$ at time $t$ is defined as

$$UCB(i, t) = \hat{\mu}_i(t) + \sqrt{\frac{2 \ln t}{n_i(t)}} \tag{5}$$

where $\hat{\mu}_i(t)$ is the empirical mean reward of arm $i$ up to time $t$, and $n_i(t)$ is the number of times arm $i$ has been played up to time $t$. We have taken $c = 2$ as the exploration parameter for similarity.

As shown in Eq. (2), the empirical mean reward $\hat{\mu}_i(t)$ for arm $i$ at time $t$ is calculated by averaging the rewards received from the arm up to time $t$. Therefore, we can define the regret at time $t$ as not playing the optimal arm $m^*$ is the difference in expected reward:

$$R(t) = \mu^* - \mu_{I_t} \tag{6}$$

where $I_t$ is the arm played at time $t$. The cumulative regret after $T$ plays is:

$$R(T) = \sum_{t=1}^{T} \left( \mu^* - \mu_{I_t} \right). \tag{7}$$

In order to bound the probability

$$P\left( \hat{\mu}_i(t) + \sqrt{\frac{2 \ln t}{n_i(t)}} \geq \mu^* \right) \tag{8}$$

we apply Hoeffding's inequality. Hoeffding's inequality provides a way to bound the probability that the sum of bounded independent random variables deviates from its expected value. Given that the rewards of each arm are bounded between 0 and 1 (as per Assumption 2), and considering the rewards as independent random variables, Hoeffding's inequality states that for any $\epsilon > 0$:

$$P\left( \hat{\mu}_i(t) - \mu_i \geq \epsilon \right) \leq \exp\left( -2\epsilon^2 n_i(t) \right). \tag{9}$$

Setting $\epsilon = \sqrt{\frac{2 \ln t}{n_i(t)}}$, we get

$$P\left( \hat{\mu}_i(t) - \mu_i \geq \sqrt{\frac{2 \ln t}{n_i(t)}} \right) \leq \exp(-4 \ln t) = t^{-4} \tag{10}$$

However, our goal is to bound the probability that the UCB of arm $i$ exceeds the mean reward of the optimal arm $m^*$. By definition of the optimal arm, $\mu_i < \mu^*$ for all $i \neq m^*$. Hence, we have

$$P\left( \hat{\mu}_i(t) + \sqrt{\frac{2 \ln t}{n_i(t)}} \geq \mu^* \right) \leq P\left( \hat{\mu}_i(t) + \sqrt{\frac{2 \ln t}{n_i(t)}} \geq \mu_i + (\mu^* - \mu_i) \right) \tag{11}$$

Since $\mu^* - \mu_i$ is a positive quantity, we can further bound this probability

$$P\left( \hat{\mu}_i(t) + \sqrt{\frac{2 \ln t}{n_i(t)}} \geq \mu^* \right) \leq P\left( \hat{\mu}_i(t) - \mu_i \geq \sqrt{\frac{2 \ln t}{n_i(t)}} \right) \leq t^{-4} \tag{12}$$

which implies that the likelihood of a sub-optimal arm appearing more appealing than the optimal arm diminishes rapidly with more trials, specifically at a rate proportional to $t^{-4}$. This steep decline in probability is a key factor in ensuring that the cumulative regret incurred by the UCB algorithm is logarithmically bounded over time. Integrating over all possible plays, the cumulative regret is bounded by:

$$R(T) \leq \sum_{i \neq m^*} O(\log T) = O(\log T). \tag{13}$$

$\square$

The above theorem substantiates the efficiency of the UCB algorithm in exploring and exploiting the mixing ratios within the ROAD framework. It highlights UCB's capability to minimize regret logarithmically, which is particularly relevant in applications with significant exploration costs, such as online learning and adaptive decision-making systems. The logarithmic regret is a cornerstone feature, ensuring that sub-optimal mixing ratios diminish over time, guiding the system toward more optimal decisions as more knowledge is accumulated.

# F COMPARISON OF ROAD WITH BALANCED REPLAY

In this section, we compare ROAD, our proposed framework, with the balanced replay approach for offline data utilization. We assess their performances in various environments and offline datasets, analyzing the underlying reasons for their differing adaptivity capabilities and the advantages inherent in ROAD.

## F.1 OFFLINE DATA UTILIZATION WITH BALANCED REPLAY

Balanced replay (BR), an approach introduced in previous work, enables the safe utilization of online samples by leveraging relevant, near on-policy offline samples (Lee et al., 2022). This strategy broadens the sampling distribution for updates around on-policy samples, facilitating timely value propagation. The primary challenge lies in designing a system capable of identifying and retrieving such relevant, near-onpolicy samples from potentially vast offline datasets. To this end, BR assess the "online-ness" of all available samples, prioritizing them based on this metric. Specifically, when updating the agent, BR samples a transition $(s, a, s') \in \mathcal{B}^{\text{off}} \cup \mathcal{B}^{\text{on}}$ with a probability proportional to the density ratio $w(s, a) := d^{\text{on}}(s, a)/d^{\text{off}}(s, a)$. This approach enables the agent to retrieve relevant, near-on-policy samples $(s, a, s') \in \mathcal{B}^{\text{off}}$ by identifying transitions with a high density ratio $w(s, a)$. However, estimating the likelihoods $d^{\text{off}}(s, a)$ and $d^{\text{on}}(s, a)$ is complex due to their potential representation as stationary distributions of intricate policy mixtures. To circumvent this, BR employs a likelihood-free density ratio estimation method, training a network $w_\psi(s, a)$ parametrized by $\psi$, using samples exclusively from $\mathcal{B}^{\text{off}}$ and $\mathcal{B}^{\text{on}}$. The training specifics are detailed in Lee et al. (2022). Previous studies have demonstrated that balanced replay can successfully enhance offline-to-online RL algorithms' sampling strategies during the online RL phase, making the online learning process more efficient in some environments.

## F.2 COMPARING ROAD WITH BALANCED REPLAY AND ANALYSIS

In this part, we juxtapose ROAD with BR to compare their offline data utilization performance across varying environmental complexities and offline dataset qualities. Specifically, our implementation of BR remains consistent with its original paper, including aspects like network architecture and loss functions. The results of our experiments are depicted in Figure 15.

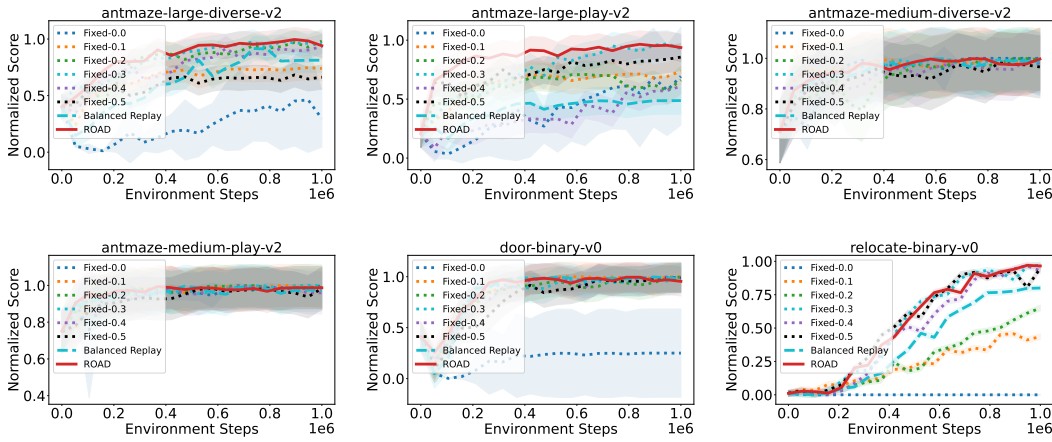

Figure 15: Normalized return curves of fixed mixing ratios, balanced replay and ROAD in AntMaze large and medium tasks (diverse and play) and Adroit door and relocate tasks, using Cal-QL (Nakamoto et al., 2023) as the backbone.

Our findings indicate that BR outperforms certain fixed mixing ratios, achieving commendable results, especially in tasks of relatively lower difficulty. Both BR and ROAD exhibit strong performance in these scenarios. However, it's important to note that there are instances where some fixed mixing ratios yield better outcomes than BR. In contrast, ROAD more effectively matches or even surpasses the optimal fixed mixing ratios. This highlights ROAD's superior adaptability in offline

data replay across diverse environments. Moreover, a significant consideration is that BR requires training an additional neural network to estimate the density ratios of different samples. This incorporation of extra network parameters introduces greater instability in the training process and increases computational requirements. In our experiments, we observed that BR demands longer training durations to complete training for the same amount of environment steps as compared to other methods. This aspect underscores the efficiency and practicality of ROAD, which achieves higher adaptability in offline data replay without the added complexity and computational overhead of an additional neural network.

## G    EXAMINATION OF ROAD WITH PEX

To further investigate the integration capability of ROAD with more recent offline-to-online RL algorithms, we dedicate this section to exploring the combination of ROAD with the policy expansion (PEX) algorithm(Zhang et al., 2023).

### G.1    THE POLICY EXPANSION APPROACH

Policy expansion (PEX)(Zhang et al., 2023) is introduced as a solution to bridge the gap between offline and online RL. The key idea is to expand the policy set after learning the offline policy. Instead of fine-tuning the offline policy directly, which risks degrading learned behaviors, PEX adds a new learnable policy to the set for online learning. The original offline policy is retained and both policies are adaptively composed for interaction with the environment. This method aims to preserve the useful behaviors learned offline while facilitating exploration and learning online. Since PEX is orthogonal to specific offline or online RL algorithms, we chose to integrate it with the CQL algorithm, mirroring the approach in the original PEX paper. This setup allows us to scrutinize the performance of PEX across different mixing ratios and its combined effectiveness with ROAD. In the initial implementation of the PEX algorithm, the mixing ratio was fixed at 0.5. To extend this analysis, we selected a range of fixed mixing ratios, specifically {0.0, 0.1, 0.2, 0.3, 0.4, 0.5}, for our evaluation. This approach enables a comprehensive examination of PEX's performance dynamics under varying degrees of offline-to-online data integration. By systematically testing these distinct mixing ratios, we aim to uncover insights into the optimal balance between offline and online data that maximizes the efficacy of the PEX algorithm. Additionally, the integration of ROAD provides a deeper understanding of how adaptive data replay strategies can enhance or complement the PEX framework.

### G.2    EXAMINING ROAD WITH PEX

In this part of our study, we conducted experiments in the HalfCheetah environment of Mujoco, incorporating three datasets of varying quality: random, medium-replay, and medium. For each dataset setting, we implemented versions of PEX with different mixing ratios, as well as versions that combine PEX with ROAD, and reported the results across four repeated experiments. The outcomes are presented in Figure 16.

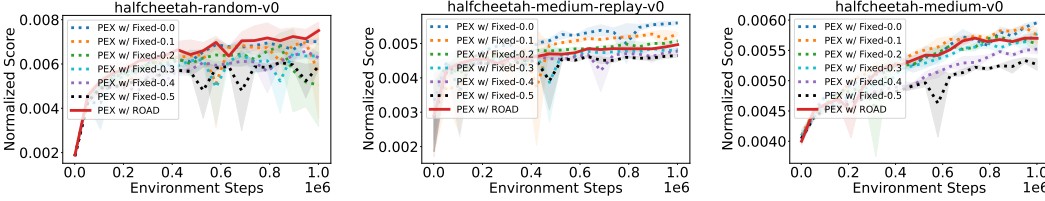

Figure 16: Normalized return curves of differnet fixed mixing ratios with PEX and ROAD with PEX(Zhang et al., 2023).

Our results reveal notable variations in the effectiveness of PEX under different mixing ratios across datasets of varying quality. Interestingly, in many trials, PEX appears to favor the use of less offline data, which deviates from the patterns observed with Cal-QL and CQL. Under these circumstances, ROAD essentially matches the optimal mixing ratio results in the random and medium datasets.

However, in the medium-replay dataset, ROAD's performance still shows some discrepancy from the optimal mixing ratio, suggesting potential room for improvement through further parameter tuning (in our current experiments, we uniformly chose $c = 1.0$ across all environments and algorithms). Moving forward, we plan to conduct experiments in a wider range of environments to facilitate a more comprehensive analysis. This broader testing will allow us to better understand the dynamics of mixing ratios and the adaptability of ROAD in conjunction with PEX across diverse scenarios. Such extensive experimentation will provide deeper insights into the interplay between offline and online data in reinforcement learning and guide future developments in this field.

