# OpenReview forum: "Adaptive Offline Data Replay in Offline-to-Online Reinforcement Learning"
_ICLR.cc/2024/Conference — ICLR 2024 Conference Withdrawn Submission_

### Official Review · Reviewer_nzb6 · 2023-10-31

**Soundness:** 2 fair
**Presentation:** 3 good
**Contribution:** 2 fair
**Rating:** 5
**Confidence:** 4

**Summary:**

The paper introduces a method named ROAD, which leverages the CQL algorithm and focuses on efficient offline data replay during the online training phase of offline-to-online RL. A crucial component of ROAD is the utilization of UCB to balance exploration and exploitation across various mixing ratios. A weighting parameter, denoted as 'c', is assigned to the exploration term in UCB calculations. Lower weights encourage the algorithm to converge after a certain level of exploration, while higher weights facilitate more extensive exploration between mixing ratios. This method is straightforward and can be easily integrated with existing offline-to-online RL algorithms.

**Strengths:**

1. The exploration of the offline-to-online problem in this study holds great relevance and is imperative for practical implementations, aligning seamlessly with the demands of real-world situations.
2. The approach taken in this paper for addressing offline-to-online RL is quite promising. The study focuses on enhancing the performance of the online phase by investigating the ratio of offline to online data. This method of improving the data ratio is highly versatile and can be applied in conjunction with any offline-to-online RL algorithm. It holds valuable insights for the research field by providing a generalizable means of improving performance.

**Weaknesses:**

1. The ROAD method introduced in this paper aims to enhance the performance of offline-to-online RL algorithms from a data utilization perspective. However, as noted in related works, approaches like Balanced Replay[1] and APL[2] share similar starting points. Balanced Replay, for instance, independently trains a neural network to control the ratio of offline and online data, making it an essential baseline for evaluating the proposed method. Nonetheless, the experimental section lacks a comparison with Balanced Replay's performance. Therefore, it is recommended that the authors include this baseline in their evaluation.
2. While the experimental section of this paper covers a variety of tasks, including AntMaze, Adroit, and Kitchen, it is important to note that there are only six specific datasets utilized. The limited coverage of dataset types in the experiments raises concerns about the overall persuasiveness of the results. For instance, questions arise as to why AntMaze only includes the "large" dataset and lacks the "medium" one, or why Adroit features "door" and "relocate" datasets but omits "pen" and "hammer." The authors could choose to focus on specific tasks, but it is advisable to conduct experiments across all dataset types for a given task, rather than selecting a few datasets for showcasing results. Furthermore, MuJoCo is a widely recognized benchmark in the offline-to-online research domain. Conducting experiments on classic tasks like HalfCheetah, Walker2d, and Hopper can facilitate comparisons with traditional methods. Therefore, it is recommended that the authors consider including experiments on these tasks as part of their study.
3. While the ROAD method is stated to be compatible with various offline-to-online RL algorithms, it is worth noting that the experimental section primarily focuses on Cal-QL, which is the sole offline-to-online RL algorithm tested. In contrast, CQL is a purely offline reinforcement learning algorithm. To demonstrate the versatility and applicability of the ROAD method, it may be beneficial for the authors to consider including experiments involving 1-2 other offline-to-online RL methods, such as AWAC[3], PEX[4], and similar approaches. Alternatively, the authors could replace the data utilization network in Balanced Replay with the ROAD method to make the ROAD results more compelling and indicative of its broader utility.

[1] Lee S, Seo Y, Lee K, et al. Offline-to-online reinforcement learning via balanced replay and pessimistic q-ensemble[C]//Conference on Robot Learning. PMLR, 2022: 1702-1712.

[2] Zheng H, Luo X, Wei P, et al. Adaptive policy learning for offline-to-online reinforcement learning[J]. arXiv preprint arXiv:2303.07693, 2023.

[3] Nair A, Gupta A, Dalal M, et al. Awac: Accelerating online reinforcement learning with offline datasets[J]. arXiv preprint arXiv:2006.09359, 2020.

[4] Zhang H, Xu W, Yu H. Policy Expansion for Bridging Offline-to-Online Reinforcement Learning[J]. arXiv preprint arXiv:2302.00935, 2023.

**Questions:**

See weakness part.

---

> ### Author Response · Authors · 2023-11-20
> **Response to Reviewer nzb6**
>
> We sincerely appreciate your thorough review and insightful feedback on our manuscript. We are encouraged by your recognition of our work's relevance to real-world situations and the promising nature of our approach.
>
> **(Weakness 1) Comparison with other data utilization approaches**
>
> We thank you for highlighting the significance of other data utilization approaches. In our revised paper, we have included an expanded discussion on related works and additional comparative experiments. Specifically, in Appendix F, we provide a comparison and analysis against the Balanced Replay (BR) method. BR predicts the density ratio between offline and online samples through a neural network and employs a prioritized sampling strategy for training. Our results, as presented in Appendix F, showcase ROAD's superiority over BR in various environments when integrated with Cal-QL. While BR shows adaptability across different tasks, ROAD consistently outperforms it, affirming the efficacy of our adaptive offline data replay in diverse settings. Moreover, ROAD’s streamlined design, requiring only a multi-armed bandit model as opposed to BR’s additional neural network, demonstrates its advantage in achieving superior adaptive performance with a more concise approach.
>
> **(Weakness 2) Coverage of a wider variety of tasks**
>
> Your suggestion to extend our experiments across a broader range of environments is well-taken. In the experiments, we initially focused on more challenging variants, such as "large" in AntMaze, to rigorously test the combination of offline-to-online algorithms with ROAD. Following your valuable feedback, we have now included results for ROAD on "medium-diverse" and "medium-replay" in AntMaze and the "pen" environment in Adroit, as detailed in the updated Appendix D. Due to time constraints, comprehensive results across MuJoCo environments, including comparisons with classic tasks, will be further elaborated in the camera-ready version. This will provide a more persuasive and complete set of findings. Thank you once again for your valuable suggestions.
>
> **(Weakness 3) Examination of ROAD with other offline-to-online algorithms**
>
> We appreciate your advice on experimenting with a wider range of offline-to-online RL algorithms. Initially, we selected Cal-QL and CQL + online finetuning for their exemplary performance in the field. Recognizing the importance of demonstrating ROAD’s compatibility with various algorithms, we have incorporated experiments with the PEX algorithm in the revised paper. These additional results and analyses are presented in Appendix G. This expansion of our experimental scope further illustrates ROAD's adaptability and performance across varying environmental complexities and offline data qualities. In our upcoming camera-ready version, we will enhance our presentation by including more comprehensive experiments with PEX and other algorithms, along with comparisons to works like Balanced Replay.
>
> Once again, we thank you for your constructive feedback and hope our revisions adequately address your concerns.

---

> > ### Comment · Reviewer_nzb6 · 2023-11-22
> >
> > Thanks for your work on the new experiments. Most of my concerns are addressed. I decided  to raise my score to 5. However, the lack of comprehensive results across MuJoCo environments hinders me from raising the score to a positive one.

---

### Official Review · Reviewer_27rC · 2023-11-01

**Soundness:** 2 fair
**Presentation:** 2 fair
**Contribution:** 2 fair
**Rating:** 3
**Confidence:** 4

**Summary:**

This paper considers the problem of using offline datasets to improve online RL. They propose an adaptive mixing ratio method where the training adaptively chooses a mixture of offline datasets and online data to train a model. The adaptivity is obtained via using bandits.

**Strengths:**

The idea of using bandits to adaptively learn a mixing balance seems novel and interesting.

**Weaknesses:**

* It is unclear to me why the reward function  $R[m]$ in Eq (2) makes sense to balance the mixing ratio. What is the interpretation of that reward signal? What would be an optimal mixing ratio?
* It's unclear how this method would work. Is there any guarantee that this bandit design would lead to some notion of optimal mixing ratio between the offline dataset and the online data?
* What is an optimal choice fo the exploration parameter $c$?
* Though the idea is interesting, the framework is quite simple and not convincing enough how this idea would work (see above questions)

**Questions:**

Please see my questions above in the Weaknesses.

---

> ### Author Response · Authors · 2023-11-20
> **Response to Reviewer 27rC**
>
> We would like to thank the reviewer for their time in reviewing our submission. We appreciate the acknowledgment that our idea is "novel and interesting".
>
> **(Weakness 1) Explanations on the reward signal and the optimal mixing ratio**
>
> The reward signal $R[m]$ in Eq (2) represents the mean reward of the agent's trajectory under a given mixing ratio $m$. This metric is pivotal for assessing various mixing ratios because (1) it directly reflects the agent’s learning efficiency in online learning via the total reward in the trajectory evaluation, and (2) during online learning, though overall trajectory rewards under different mixing ratios tend to increase as the agent's ability improves, a beneficial mixing ratio will still emerge superior among the candidates. This superiority encourages a balance between offline demonstration and online exploration. An optimal mixing ratio is not necessarily static; it can evolve with the agent's learning progress. For instance, a higher ratio may be preferable early in the learning phase to leverage high-quality offline data, whereas a lower ratio might be more beneficial later to encourage online exploitation.
>
> **(Weakness 2) Optimality guarantee of ROAD**
>
> We appreciate your suggestion. In Appendix E, we provide additional proofs under the assumption of an existing optimal mixing ratio $m^*$. ROAD's adaptive mixing ratio selection has certain optimality guarantees as detailed there. Moreover, in practical scenarios, the optimal mixing ratio often varies in response to different environments, offline dataset qualities, and the agent's training progress. Our experiments demonstrate this adaptiveness, where ROAD achieves superior performance across various environments and dataset qualities, in conjunction with multiple offline-to-online RL algorithms (please refer to Appendices D, F, and G in the updated paper).
>
> **(Weakness 3) Optimal choice of exploration parameter**
>
> In the experiments of Figures 3 and 4, we implemented ROAD with $c=1$, as stated in Section 5.3. We have also supplemented Section 5.1 with a rationale for selecting the exploration parameter. Section 5.3's ablation study showcases ROAD's robustness across different exploration parameters, indicating its adaptive capability for offline data replay.
>
> **(Weakness 4) Further discussion on the idea of ROAD**
>
> ROAD's design leverages online feedback from the agent to evaluate the impact of different mixing ratios on online learning. By integrating a bandit strategy for evaluation and exploitation across these ratios, ROAD dynamically adjusts offline data replay, enhancing the agent's online learning performance. We have rigorously evaluated ROAD's adaptive mixing ratio selection capabilities across diverse task difficulties and offline data qualities. The experimental results underscore its effective adaptability. ROAD, when integrated with various offline-to-online RL algorithms, has shown promising results in multiple environments and offline datasets (refer to newly added Appendices D, F, and G). Comparative analyses with variants further substantiate the rationale behind adopting the UCB algorithm in ROAD (see Appendix B). Furthermore, we have provided proof of optimality guarantee, demonstrating ROAD’s effectiveness in converging to an optimal mixing ratio (refer to Appendix E). We humbly request that our explanations, theoretical foundations, and empirical validations be considered in reassessing the design and implementation of this idea.

---

### Official Review · Reviewer_jaY3 · 2023-11-01

**Soundness:** 3 good
**Presentation:** 2 fair
**Contribution:** 2 fair
**Rating:** 5
**Confidence:** 4

**Summary:**

This paper addresses offline-to-online RL, where an RL agent is pre-trained with an offline dataset and then fine-tuned in an online environment. While most existing works use a fixed mixture ratio of offline and online data during this fine-tuning stage, this study proposes an adaptive adjustment of this ratio. The selection is driven by a bandit algorithm that responds to environmental returns.

**Strengths:**

- The core idea is intuitive and appears to be effective, as supported by the empirical results.

-  The empirical evaluation is comprehensive.

**Weaknesses:**

- The details of implementation are somewhat unclear (see the questions section).

- While ROAD tunes exploration weight $c$, a comparison with fixed-ratio baseline with the optimal ratio $m^*$ would be informative, where the optimal ratio $m^*$ should be selected in a manner similar to selecting $c$.

- Lacking comparison to Balanced Replay (BR) [1]: Given that both ROAD and BR aim to adjust the offline/online data ratio as replay buffer components, a comparison would offer valuable insights into ROAD's effectiveness.

[1] Lee, Seunghyun, et al. "Offline-to-online reinforcement learning via balanced replay and pessimistic q-ensemble." Conference on Robot Learning. PMLR, 2022.

**Questions:**

- In Figure 5, it seems the bandit learner never explores the arm with $m=0.5$.

- What value of exploration weight parameter $c$ corresponds to the curves shown in Figure 4?

- How frequently was $m$ updated?

- Algorithm 1 appears to be different from standard practices where data collection and gradient updates are done simultaneously. I'd like to confirm with the authors the training process of ROAD. Is it accurate that ROAD proceeds by: (1) collecting $n$ trajectories (equivalent to $n*T$ environmental steps, with $T$ as the trajectory length), (2) updating the bandit learner and $m$ using these $n$ trajectories, and then (3) then running $k$ gradient steps?

- (Continued) If my understanding of the algorithm is correct, I'm curious to know if this modification was extended to Cal-QL for the fixed ratio baselines, since such changes in the training process might non-trivially impact offline-to-online performance.

---

> ### Author Response · Authors · 2023-11-20
> **Response to Reviewer jaY3**
>
> We appreciate your thorough review of our work and your insightful comments. We are encouraged that you found that "the core idea is intuitive" and "the empirical evaluation is comprehensive".
>
> **(Weakness 2, Question 2) Comparison with optimal mixing ratio $m^\star$**
>
> Regarding the choice of exploration weight in ROAD, $c=1$ in all experiments presented in Figures 3 and 4, as clarified in Section 5.3. We have now elaborated on the selection of the exploration weight in Section 5.1. We value your suggestion on comparing with the optimal fixed mixing ratio $m^*$. Accordingly, we have added Appendix D.2, which details the performance of different fixed mixing ratios across various environments and assesses the optimal ratio $m^*$ using a similar methodology to our approach for selecting $c$. This comparison demonstrates ROAD's consistent advantage over the optimal fixed ratio, highlighting its adaptability across diverse environments, beyond the constraints of a fixed mixing ratio. We invite you to review the detailed comparisons and analyses in Appendix D.2.
>
> **(Weakness 3) Comparison to balanced replay**
>
> We are thankful for your suggestion to compare with Balanced Replay (BR). BR employs a neural network to predict the density ratio of different samples, including offline and online data, using this priority strategy for training sample selection. In Appendix F, we now provide a comparison of ROAD, based on Cal-QL, with BR in multiple environments. While BR shows improvements over certain fixed mixing ratios, indicating some adaptability, ROAD demonstrates superior results. Additionally, BR's reliance on an additional neural network for training and inference increases the overall parameter count and computational load. In contrast, ROAD's adaptive selection utilizes a more concise multi-armed bandit model, achieving better adaptive performance with a simpler design.
>
> **(Question 1, Question 3) Exploration and update of different arms**
>
> In Figure 5, we illustrate the mean and standard deviation of ROAD's choices of mixing ratio over four repeated experiments, across different tasks, over time. ROAD does explore different arms, providing a more accurate estimation of the value of each arm (mixing ratio). The update process of mixing ratio in ROAD, as detailed in line 8 of Algorithm 1, involves using the updated $R_m$ and $N_m$ for each selection of a new arm. We have now provided further clarification in the paper.
>
> **(Question 4, Weakness 1) Data collection and gradient updates in ROAD**
>
> We would like to clarify that the data collection and gradient update processes in ROAD are consistent with those in algorithms like Cal-QL. When integrated with other algorithms, ROAD adapts seamlessly to various offline-to-online RL frameworks. Like Cal-QL, ROAD involves collecting one or more trajectories into the replay buffer before sampling from it for gradient updates. The bandit learner update in ROAD uses only the trajectory rewards, with other parameters (such as the number of trajectories $n$ and the number of gradient steps $k$ ) being consistent with the implementations of Cal-QL and CQL. This ensures consistency with fixed mixing ratio baselines in our experiments.
>
> We hope these clarifications address your concerns, and we thank you again for your valuable feedback.

---

### Official Review · Reviewer_nGv3 · 2023-11-04

**Soundness:** 2 fair
**Presentation:** 2 fair
**Contribution:** 3 good
**Rating:** 5
**Confidence:** 3

**Summary:**

Most offline-to-online RL requires mixing the offline data with online data based on some mixing ratio. Prior work most focus on the static ratios, however, the optimal ratio might be a function of task complexity, offline data quality, task environment, etc. This paper tackles this problem by studying an adaptive way to update the mixing ratio through the exploration reward in the online stage. It is being framed as a multi-arm bandit problem, with the action space as the possible mixing ratios, the reward is the average trajectory reward, and the goal is to find the optimal mixing ratio. It then utilizes the UCB algorithm here and shows promising empirical results.

**Strengths:**

- The paper points out that the mixing ratio for offline and online data is task dependent, and provides some empirical results to demonstrate. The study in this paper is very well motivated, i.e., the mixing ratio should be found in a task-dependent and adaptive manner.

- It nicely frames this as a bandit problem, and utilizes the UCB approach, which shows promising empirical results.

- Beyond simple comparison with the baselines, the paper also does some investigation of the methods, such as how the adaptive ratio changes as training continues? could it automatically find something near the optimal ratio? as well as some ablation studies regarding the hyper-parameter used in the bandit algorithms, this provides deeper understanding of the method.

**Weaknesses:**

- One thing I found not super clear is the method section of the paper, which is the most important part of the paper, i.e., Section 4.1. Equation 2 is a little bit confusing, what do you mean by value estimate of $m$. If I understand correctly, we could simply using the average trajectory reward under mixing ratio $m$ as the reward, I am not super how Equation 2 comes. Some notations are used w.o. definition, such as sum(N) in Equation 3. The method part should be introduced more rigorously and clearly.

- One motivation about the paper is that the mixing ratio is task dependent, environment dependent as well as time-dependent (as training goes). After proposing the ROAD, it would be great to revisit this question, i.e., to establish the connection of the mixing ratio with some notion of task difficulty, etc, either empirically or theoretically. Section 5.4 is an example of this in terms of data quality, but it would be great to investigate other dimensions as well.

- Given the paper is utilizing bandit algorithms here, more bandit algorithms such as Thompson Sampling could be tested here as well. It would also help to improve the paper if there is some optimality guarantee of the proposed algorithm.

**Questions:**

See weakness section.

---

> ### Author Response · Authors · 2023-11-20
> **Response to Reviewer nGv3**
>
> We express our sincere gratitude to the reviewer for your insightful feedback. We are encouraged by the acknowledgment that our paper is "very well motivated" and that our experiments demonstrate "promising results."
>
> **(Weakness 1) Clarification of the method section**
>
> Thank you for highlighting ambiguities in the method section. As you rightly interpreted, we indeed use the average trajectory reward under mixing ratio $m$ as the reward. Equation 2 represents an incremental online updating approach (Cumulative Moving Average) for this purpose. The motivation behind this updating style is the substantial sample requirement in online RL phases. Incremental updating, maintaining only the current average and count of data points, reduces overall costs. In Equation 3, the term sum(N) refers to the total number of times all arms have been pulled. We appreciate your suggestions for refining the method section. We have now polished it to make the logic clearer and corrected the issue of notations used without definitions.
>
> **(Weakness 2) Revisiting the motivation after proposing ROAD**
>
> We are grateful for your suggestion to revisit our motivation following the introduction of ROAD and for recognizing our analysis of offline data quality. To further explore the tendencies of the mixing ratio and ROAD's ability for adaptive data mixing across varying task difficulties, we utilized the D4RL's antmaze series environments. Please refer to the newly added Appendix D, where we compare the performance of various baselines and ROAD in environments of different complexities, further analyzing ROAD's adaptive capabilities. Future versions of our paper will include extensive experiments across a more diverse range of environmental difficulties and data qualities.
>
> **(Weakness 3) Exploring More Bandit Algorithms and Optimality Guarantee**
>
> In ROAD, we employed the UCB algorithm for adaptive offline data replay through a multi-armed bandit framework. To assess the adaptability and performance of UCB in the context of mixing ratio selection, we evaluated potential enhancements, including discounted UCB and sliding-window UCB, in Appendix B. UCB demonstrated considerable adaptability, leading to its implementation in ROAD for adaptive offline data replay. Indeed, other bandit algorithms, such as Thompson Sampling, might exhibit different adaptability across tasks and offer potential improvements within a similar framework to ROAD. This represents a promising avenue for future work. Furthermore, regarding the convergence of the multi-armed bandit in offline data replay within ROAD, we now provide a proof in Appendix E. Assuming the existence of an optimal mixing ratio $m^*$ (as empirically observed in each environment), ROAD assures an $O(\log T)$ regret. Our diverse experimental results further corroborate ROAD's adaptive selection across varying task difficulties, offline data qualities, and training times.